# From the Perspectives of Pollution Governance and Public Health: A Research of China’s Fiscal Expenditure on Energy Conservation and Environmental Protection

**DOI:** 10.3390/ijerph20116018

**Published:** 2023-05-31

**Authors:** Di Zhang, Xiao Dong

**Affiliations:** Department of Environmental Science and Engineering, Fudan University, Shanghai 200438, China

**Keywords:** financial expenditure, pollution governance, public health, data envelopment analysis

## Abstract

Improving the scale and effectiveness of China’s energy conservation and environmental protection fiscal expenditure is crucial to enhancing the capacity of ecological and environmental governance of China, considering the dual perspectives of pollution governance and public health. This article first explains the mechanism by which national energy conservation and environmental protection fiscal expenditure can improve pollution control and promote public health. Secondly, this article scrutinizes the current status and limitations of China’s fiscal expenditure, highlighting the contribution of fiscal expenditure in the construction of ecological civilization from the standpoints of environmental governance and public health. Additionally, this study empirically uses DEA to measure the efficiency of the government’s fiscal expenditure. Conclusions found that: First, environmental protection fiscal expenditure is mainly focused on technological transformation and pollution control, while relatively little is spent on public health protection. Second, the efficiency of environmental protection fiscal funds is relatively low. These suggestions aim to optimize the positive impact of energy conservation and environmental protection fiscal expenditure for improving pollution governance and promoting public health.

## 1. Introduction

In recent years, with the continuous development of China’s economy, environmental problems have become increasingly prominent. Improving pollution governance and creating a healthy ecological environment is a significant issue that concerns the nation’s fate and the people’s well-being. It has proved that China’s past economic development model of “treatment after pollution” has led to the accumulation of environmental problems over the years, which has laid hidden dangers for sustainable development. China’s economy is facing multiple challenges, such as speed shift and structural adjustment, and the environmental load has reached a critical level. Environmental problems such as hazy weather, water pollution, and excessive heavy metals in the soil have emerged frequently, putting enormous pressure on ecological restoration efforts and seriously endangering public health. The damage to public health caused by environmental problems is characterized by complexity, long-term and irreversibility. Coupled with the transformation of human lifestyles under the influence of economic, scientific and technological development, public health has deepened and becomes more complicated by the force of environmental factors. For example, the air in cities is polluted by vehicle exhaust and industrial fumes, which has led to a rapid increase in the number of respiratory diseases and a significant increase in the incidence of allergic diseases; the introduction and application of transgenic technologies implicitly affect human genetics; children’s hearing and attention to learning are disturbed by noise pollution, which not only affects the health of modern people but also threatens future generations.

To transform unsustainable development, promote ecological environment improvement and resource conservation, and enhance environmental health, China has been paying more attention to environmental issues. In recent years, with the improvement of citizens’ living standards and increasing attention to human health, environmental issues have received unprecedented attention. Therefore, environmental protection and governance in China should focus on solving the outstanding environmental problems that harm the public’s health. The environment and public health are related to the vital interests of the people. In order to avoid environmental problems from affecting social and economic development, relevant departments should properly prevent and deal with environmental damage that damages public health. The focus of this paper is to investigate the impact of energy conservation and environmental protection fiscal expenditure on public health from a sustainable development perspective. Specifically, this study aims to examine China’s fiscal expenditure concerning its role in promoting the well-being and safety of the population. Furthermore, this study explores how energy conservation and environmental protection fiscal expenditure can support and optimize efforts to strengthen environmental pollution governance and improve public health. In doing so, this paper underscores the importance of sustainable development principles in promoting the health and safety of people in China and beyond. Additionally, this report draws on a range of scholarly sources to critically analyze the current status and limitations of China’s energy saving and environmental protection fiscal expenditure, as well as provide recommendations for improving the management and effectiveness of these funds. Overall, this paper seeks to contribute to ongoing discussions and debates on the relationship between environmental protection, public health and sustainable development.

## 2. Literature Review

### 2.1. Study on the Pollution Control Effects of Environmental Fiscal Expenditure

There are few studies in the existing literature exploring the environmental governance effects of environmental fiscal expenditure, where most scholars believe that environmental fiscal expenditure has an abatement effect on environmental pollution. For example, Brock et al. (2004) [1] and Galinato (2017) [2] introduced the environment as a factor in the endogenous growth model. The empirical results show that an increase in environmental fiscal expenditure could significantly reduce emissions of environmental pollutants and improve environmental quality. Using three-panel models, Zhang Y (2014) [3] empirically analyzed the relationship between environmental fiscal expenditure on three pollutants, and the empirical results showed that environmental fiscal expenditure had a significant inhibitory effect on the emissions of these three pollutants. Reaching similar conclusions, Yang et al. (2016) [4] analyzed the relationship between environmental protection fiscal expenditure and environmental quality in Liaoning Province. The empirical evidence showed that the growth of environmental protection fiscal expenditure could significantly improve environmental quality. Additionally, in the environmental governance of environmental protection fiscal expenditure, Xiaohui Z et al. (2017) [5] combined the proportion of fiscal expenditure to GDP and the environmental pollution index in each province. It divided China’s 30 provinces into 2 regions with more substantial and weaker pollution governance effects of environmental protection fiscal expenditure. The empirical analysis found that China’s environmental protection fiscal expenditure has a significant inhibitory effect on the emission of environmental pollutants and a significant governance effect.

However, some scholars have come to different conclusions. For example, Bergh (2010) [6] argues that although environmental protection fiscal expenditure has a better effect on the protection of the environment and pollution control, the increase in the proportion of environmental protection fiscal expenditure in total fiscal expenditure will squeeze the balance of other fiscal expenditure items. Therefore, environmental protection and governance should start by improving the efficiency of environmental protection fiscal expenditure. Dong C.C. (2015) [7] takes five provinces in Northwest China as the research object. Through an empirical study of relevant regional data, he found that environmental protection fiscal expenditure can significantly suppress industrial wastewater and sulfur dioxide emissions. Still, the suppression effect on industrial solid waste emissions is not significant. Xiao J.Y (2018) [8] used panel data from 30 Chinese provinces selected from 2003 to 2013. The empirical results showed that there was a significant threshold effect between government environmental management policies and several indicators measuring wastewater quality, among which, for the proportion of chemical oxygen demand and ammonia nitrogen in wastewater, after the per capita economic level crossed the threshold, government environmental management policies were instead this indicates that the local economic level is still an essential factor influencing environmental governance.

### 2.2. Impact of Government Environmental Policies on Public Health

The health level of the nation is not only related to people’s well-being but also to economic and social stability and sustainable development, and the potential hazards caused by environmental pollution to national health are essential reasons for the government to carry out environmental regulation. In the field of environment and health, the existing foreign research mainly focuses on two directions, one is to verify the negative impact of environmental pollution on national health, and scholars generally affirm this view. For example, Tanzi (1997) [9] found that there is a risk of accelerated depreciation in the health of people living in heavily polluted areas; Currie et al. (2009) [10] used New Jersey, USA, as a study subject and confirmed that increased carbon monoxide concentration leads to increased infant mortality. Second, we study the changes in national health brought by environmental regulations, which include both the short-term effects of certain temporary environmental policies before and after their implementation (Esty et al. 1996 [11]; Worthington et al. 2000 [12]; Benson et al. 2013 [13]; Ashenfelter and Greenstone 2002 [14]), such as the proposed legislation on sulfur-containing fuels in Hong Kong, the Beijing environmental regulation during the preparation of the Olympic Games, etc., but also the long-term effects of environmental laws and regulations on national health changes (Fare and Grosskopf, 2003 [15]; Sanders and Stoecker, 2015 [16]). The results generally suggest that environmental regulations can somewhat reduce morbidity and mortality.

In contrast to foreign studies, domestic scholars have likewise confirmed the non-negligible hazards to national health caused by environmental pollutants such as PM10, sulfur dioxide, industrial soot, and sewage (Yanqing M and Wenjing C, 2010 [17]; Chen S.Y and Chen J, 2008 [18]; Bie T and He L, 2018 [19]). For the study of environmental policies and health issues, the early domestic literature mainly focused on constructing China’s environmental and health standards system for consideration. Lu H.Y (2013) [20] and Cai L (2012) [21] argued that the establishment and implementation of environmental standards in China could not meet the actual needs of the nation for health protection, which was mainly caused by the uncoordinated internal system of environmental protection and uncoupled external support, and that a sound environmental standard and implementation tool led by health values should be established. With the deepening development of environmental policies, the health impacts they brought have gradually drawn scholars’ attention. Based on the Grossman model, Wang Bing and Li M.J (2012) [22] found that sewage discharge reduces the health level of rural middle-aged and elderly people and that the use of running water and the promotion of “toilet and water conversion” projects can significantly reduce the adverse health effects of sewage, so the government should increase its expenditure on public health.

A review of the previous literature reveals few studies on the impact of environmental fiscal expenditure on public health. The vast majority of studies indicating public health from the fiscal perspective are based on traditional biological theories, i.e., exploring from the perspective of health expenditure, including the scale of health expenditure, the degree of supply of medical service utilization, etc., ignoring the impact of other fiscal expenditure on public health. Therefore, with the development of the social economy, in addition to health expenditure, the impact of environmental protection fiscal expenditure, which is closely related to health, on public health also deserves our further in-depth analysis.

## 3. The Mechanism of the Role of Public Health and Fiscal Expenditure on Energy Conservation and Environmental Protection

Environmental issues are the top priority for China to achieve sustainable development, and public health is the primary goal of the government to carry out environmental protection and pollution governance, both of which represent the core demands of China’s ecological civilization construction in the new era, and are the direction of development for energy conservation and environmental protection fiscal expenditure to follow [23]. Based on clarifying the connotation of the government’s functions and fiscal expenditure, this section takes environmental governance and public health as the starting point to explain the mechanism of the role of fiscal expenditure on environmental protection in promoting environmental improvement and enhancing public health. It also explores how fiscal expenditure on energy conservation and environmental protection affects public health through environmental governance.

### 3.1. The Dialectical Relationship between Environment and Health

While the evolution of industrial civilization has brought great wealth to humankind, it has also led to the over-exploitation and irrational use of energy, resulting in ecological crises and pollution accidents that threaten human health [24]. China’s early traditional economic growth model led to severe environmental evils, and environmental problems emerged in stages during the industrialization process in developed countries over the last century [25]. The gap between the people’s demand for a beautiful ecological environment and the current quality of environmental public goods supply has increased rather than decreased in China, which has entered a new economic normal.

“A good ecological environment is the universal welfare of people’s livelihood” and the governance of the ecological environment should be oriented to the health problems of people’s livelihood [26]. Thus, it is clear that the ultimate focus of ecological and environmental governance lies in optimizing the environment and benefiting people’s livelihoods. The quality of human life is inextricably linked to the ecological environment [27]. A quality environment can improve residents’ life satisfaction and happiness index, health level and life expectancy [28]. On the contrary, the inhabitants of a poor living environment are deprived of continuous material security of survival, ecological rights, and physical and mental health, which makes it challenging to lead a healthy and happy life and maintain a proper life expectancy [29]. Ecological civilization is therefore committed to the people’s livelihood and their well-being. Therefore, the construction of ecological civilization is dedicated to the people’s economic prosperity and to “let the people drink clean water, breathe clean air and eat safe food”. In addition, it is necessary to let the people thoroughly enjoy the ecological benefits of social development and satisfy their good wishes for ecological affluence. These visions are pinned to both environmental and health factors.

As an economical vehicle to implement the government’s will, energy conservation and environmental protection fiscal expenditure aims to curb the decline of environmental quality and national health risks caused by pollution problems. The direction of the expenditure must be based on the macro layout of enhancing the intensity of environmental pollution governance and improving public health.

### 3.2. The Role of Energy Conservation and Environmental Protection Fiscal Expenditure on Public Health

“Environmental protection and governance should focus on solving outstanding environmental problems that harm the health of the masses” and “focus on prioritizing efforts to solve outstanding environmental problems that harm the health of the masses, such as fine particulate matter (PM2.5)”. The latest revision of the Environmental Protection Law of the People’s Republic of China has made “safeguarding public health” one of the legislative objectives, which shows that national health has become an essential direction of environmental governance. The latest revision of the Law of the People’s Republic of China on Environmental Protection has made “protection of public health” one of the legislative objectives. As the material guarantee of environmental management, the expenditure of funds must be focused on the direction of environmental protection that can improve people’s health, and the impact on health is mainly transmitted through environmental quality.

#### 3.2.1. The Mechanism of the Impact of Financial Expenditure on Energy Conservation and Environmental Protection on Physical Health

Ecological economics states that health is a product of the ecological relationship between humans and the environment [30]. The waste and unreasonable use of resources in human production and life will produce harmful factors in all aspects of the environment, such as the atmosphere, water and soil. Once their quantity, concentration and duration exceed the self-cleaning capacity of the environment, they will break the ecological balance and cause harm to human health [31].

In the face of the risks posed by environmental pollution to national health, fiscal expenditure can impact national health by “solving outstanding environmental problems that harm people’s health”. However, it does not directly contribute to improving health levels. First, fiscal expenditure on energy conservation and environmental protection can remedy environmental pollution [32]. From the content of the expenditure, it can be seen that the fiscal expenditure prevents and treats pollutants that pose direct hazards to human beings, such as water, air, solid waste, chemicals, radioactive sources and radioactive waste [33]. Additionally, the accurate pollution level can be dynamically grasped by building pollution source monitoring mechanisms and improving environmental monitoring, which effectively curbs the deteriorating trend of environmental pollution and slows down the direct infringement of environmental pollution on human health [34]. Second, energy conservation and environmental protection fiscal expenditure can promote energy conservation. In terms of expenditure content, fiscal expenditure support cleaner production, implement subsidies for renewable energy sources such as wind, solar, biomass and promote the development of the circular economy. The production of pollutants is controlled at the source to reduce the potential health risks caused by environmental pollution [35].

#### 3.2.2. The Mechanism of the Impact of Financial Expenditure on Energy Conservation and Environmental Protection on Mental Health

In addition to physical damage, environmental pollution makes humans live in a long-term uneasy environment. The spirit under the pressure of “an existential crisis” is always tense, seriously affecting people’s mental health [36]. The haze, for example, is a gloomy and unsettling environment. The lack of sunlight, low air pressure, and reduced outdoor activities caused by gloomy weather can affect people’s emotions through physiological discomfort, resulting in mental laziness, anxiety, irritability and depression [37]. Suppose this adverse environment is too intense or lasts too long. In this case, it may induce abnormalities in the higher neural activity of sensitive individuals, causing symptoms such as anxiety and depression, which threaten people’s life and health [38].

At this time, fiscal expenditure can play a positive role through ecological restoration. In terms of specific content, environmental protection fiscal expenditure takes the form of natural forest protection, returning farmland to forest, returning grazing to grass, nature reserve construction, wind and sand desert treatment, and urban greening to conserve the ecological environment [39]. The full release of microorganisms and plants in the growth process of the metabolic capacity of pollutants and the acceleration of the ecological cycle, thus creating a new natural environment, which is conducive to the development of residents’ physical and mental health.

## 4. Analysis of the Development Status of Fiscal expenditure on Energy Conservation and Environmental Protection and expenditure Efficiency in China

Environmental protection in China has undergone a complicated development process along with our economic development, which can be roughly divided into two stages. Before 2007, there was no substantial financial allocation for environmental protection expenditure and no monetary funds dedicated to environmental protection in China. Environmental protection was only attached to other financial accounts, and because the socialist planned economy influenced China, the focus of government finance was not on environmental protection. In 2007, when the reform of government revenue and expenditure classification was implemented, environmental protection expenditure (changed to energy conservation and environmental protection in 2012) was officially included in the government budget and accounts. For the first time, “211 Environmental Protection” was included in the 17 categories of functional classification of expenditure.

### 4.1. Expenditure Scale Analysis

As energy conservation and environmental protection work is being carried out, the scale of fiscal expenditure is gradually expanding. The fiscal expenditure in 2007 was RMB 99.582 billion; in 2021, the fiscal expenditure was RMB 552.514 billion. The value-added fiscal expenditure is RMB 452.932 billion, and the scale of expenditure in 2021 is 5.55 times that in 2007. Among them, expenditure reached the highest value of RMB 739.020 billion in 2019 and decreased abruptly in 2020, but the overall growth trend is maintained. The proportion of environmental protection expenditure to fiscal expenditure increases yearly with little ups and downs and begins to fall after reaching a maximum value of 3.09% in 2019. The scale and proportion of fiscal expenditure on energy conservation and environmental protection in China decreased in 2020 due to the impact of the new crown pneumonia. The statistical results are shown in Table 1. Nevertheless, overall, China is paying more and more attention to environmental protection and energy conservation. Environmental protection has become part of the government’s daily management, just like education, medical care and social security.

### 4.2. Analysis of Expenditure Structure

The government has constantly been exploring and promoting environmental protection work, and the statistical caliber of energy conservation and environmental protection expenditure has been gradually refined and improved. As of 2021, China has set up 14 sections and 62 items under the first level of expenditure, which are the second-level and third-level accounts. The detailed establishment of branches allows environmental protection funds to be implemented into specific projects.

Table 2 shows the share of central and local specific expenditure in energy conservation and environmental protection expenditure in 2021. We can see that there is not only a huge gap between the central and local financial expenditure in terms of quantity but also a big difference in the structure of the expenditure. In 2021, energy management affairs, renewable energy, natural forest protection and pollution reduction account for 40.54%, 7.05%, 6.40% and 5.07% of the expenditure on energy conservation and environmental protection, respectively. It can be seen that energy management affairs, renewable energy, natural forest protection, pollution abatement and other environmental protection work dominate the central expenditure in 2021. Furthermore, local pollution prevention and governance, energy conservation and utilization, natural ecological protection, and environmental protection management affairs accounted for 35.70%, 12.02%, 10.15% and 5.94%, respectively. This shows that local expenditure is dominated by pollution prevention and governance, natural ecological protection, energy conservation and utilization, environmental protection management affairs, and other prior protection work. This pattern of distribution between the central and local environmental protection expenditure structure corresponds to the functions of the respective governments. The central government focuses on macro-governance, grasping the overall situation and overall. In contrast, the local government considers local conditions, starting from the immediate interests of residents, starting with pollution prevention and governance with quick results, forming a top-down environmental management mechanism. Currently, China’s ecological environment has yet to achieve fundamental improvement. The government has invested much financial expenditure in pollution prevention and governance. The dynamic development trend of the government’s financial expenditure closely follows the actual environmental situation in China, and the two match each other.

### 4.3. Expenditure Efficiency Analysis

#### 4.3.1. Subsubsection

Determining scientific and reasonable evaluation methods is the basis to ensure that the evaluation of the efficiency of environmental protection financial expenditure can be carried out smoothly and is a prerequisite for designing a simple and practical evaluation index system. In selecting methods for evaluating the efficiency of environmental protection financial expenditure, most of the existing literature uses data envelopment analysis (DEA), hierarchical analysis (AHP), and cost–benefit analysis [40]. DEA is a statistical method to calculate the performance values of multiple inputs and outputs, and the calculation results are also more objective and precise.

In 1978, operations researchers A. Charnes, W.W. Cooper, and E. Rhodes first proposed the data envelopment analysis (DEA) model for evaluating relative effectiveness among the same sectors (hence the name DEA validity). Their first model was named as insert C^2^R model [41]. From the point of view of production functions, this model is a very ideal and productive way to study “production sectors” with multiple inputs, especially with numerous outputs, that are simultaneously “scale efficient” and “technology efficient”. It is an ideal and fruitful approach to study “scale efficient” and “technology efficient” production sectors with multiple inputs, especially with multiple outputs.

In this stage, the initial efficiency evaluation is performed using input–output data. The DEA models are divided into input-oriented and output-oriented, and different orientations can be chosen depending on the specific purpose of the analysis. In general, the input-oriented BCC (variable payoffs to scale) model is chosen in most studies where the three-stage DEA model is applied. For either decision unit, the BCC model in the pairwise form under input orientation can be expressed as
$$\begin{array}{l} \;min\theta -\varepsilon ({\hat{e}}^{T}S^{-}+e^{T}S^{+})\\ s.t.\left\{\begin{array}{l} \sum_{j=1}^{n}X_{j}\lambda_{j}+S^{-}=\theta X_{0}\\ \sum_{j=1}^{n}Y_{j}\lambda_{j}-S^{+}=Y_{0}\\ \lambda_{j}\geq 0,S^{-},S^{+}\geq 0 \end{array}\right. \end{array}$$

The DEA model is essentially a linear programming problem. Represents the decision unit and is the input and output vectors, respectively.

If $\theta \;=1,\;S^{+}=S^{-}=0$, then the decision cell DEA is valid.

If $\theta \;=1,\;S^{+}\neq 0,\;or\;S^{-}\neq 0$, then the decision cell weak DEA is valid.

If θ<1, then the decision unit non-DEA is valid.

The efficiency value calculated by the BCC model is the combined technical efficiency (TE), which can be further decomposed into scale efficiency (SE) and pure technical efficiency (PTE), TE = SE×TE.

#### 4.3.2. Indicator System Construction

The input and output variables in the DEA measurement are selected in this paper as follows.

Input indicators: Given that this chapter aims to examine the efficiency of fiscal expenditure in China, the per capita fiscal expenditure on energy conservation and environmental protection in China during 2007–2021 is chosen as the input indicator for the efficiency measurement.

Output indicators: Combining the main directions and contents of fiscal expenditure and considering the dimensions of the environment and human health, based on the availability of relevant data, this paper divides the output variables into three aspects.

First, is pollution governance. In order to examine the impact of environmental pollution on human health, five indicators are selected: industrial smoke (dust) emissions, industrial sulfur dioxide emissions, total industrial wastewater emissions, centralized treatment rate of sewage treatment plants, and harmless treatment rate of domestic waste, among which industrial smoke (dust) emissions, industrial sulfur dioxide emissions, and total industrial wastewater emissions are taken as non-desired outputs, and centralized treatment rate of sewage treatment plants and the harmless treatment rate of domestic waste are expected outputs.

Second, energy saving. Three indicators are selected to examine the actual extent of energy utilization: industrial electricity consumption per unit of GDP, urban water supply per unit of GDP, and total utilization rate of general industrial solid waste. Among them, industrial electricity consumption per unit of GDP and urban water supply per unit of GDP are non-desired outputs. The comprehensive utilization rate of general industrial solid waste is the desired output.

Third, ecological repair. To examine the maintenance status of natural ecology and the public’s demand for environmental health, two indicators, the green area coverage of built-up areas and the area of planted forests, are selected, both of which are taken as desired outputs.

Each input and output is shown in Table 3.

#### 4.3.3. Data Sources and Processing

The data used in this chapter on the efficiency of fiscal expenditure on environmental protection are from the China Statistical Yearbook, China Environmental Statistical Yearbook, China Energy Statistical Yearbook, China Financial Yearbook, Statistical Bulletin of National Economic and Social Development, as well as the WIEGO statistical database and national data websites for 2007–2021.

Given the feasibility and availability of the data, this paper selects the statistical data of China during 2007–2021, except for the per capita fiscal expenditure on environmental protection, which is obtained by dividing the fiscal expenditure by the average annual population. The rest of the data are directly based on the data of the corresponding Statistical Yearbook. The non-desired outputs are counted down to ensure the positivity of the data.

#### 4.3.4. Empirical Analysis of Fiscal Expenditure Efficiency Based on the DEA Method

This section employs the Deap2.1 software to measure the DEA-BCC efficiency values of China’s fiscal expenditure on energy conservation and environmental protection under variable returns to scale (VRS) conditions based on panel data on inputs and outputs from 2007 to 2021 in China. The resulting measurements include comprehensive technical efficiency (TE), pure technical efficiency (PTE), and scale efficiency (SE) [42]. Comprehensive technical efficiency is the product of pure technical efficiency and scale efficiency, used to measure the overall effectiveness of fiscal funds. As this study focuses on China, the year is selected as the decision unit for conducting a time-series DEA analysis with only one decision unit. At this point, we assume that all years have the same technical efficiency (i.e., the efficiency value is 1). The efficiency value calculated at this point represents the efficiency value relative to the entire period and can be used to compare which year had the highest efficiency. Therefore, this paper analyzes the comprehensive efficiency.

As seen from the table, the mean value of the integrated technical efficiency of fiscal expenditure on energy conservation and environmental protection in China fluctuates and then rises between 2007 and 2021, with the absolute value moving roughly in the range of 0.57–1. At the temporal level, in terms of the adequate level, there are two years in which the comprehensive technical efficiency has reached the practical level during the fifteen years, 2007 and 2021, respectively. The total efficiency value fluctuates between 0.57 and 0.65 during 2020–2019, which is a certain distance from the practical level (i.e., the efficiency value is 1).

In terms of scale efficiency, the scale efficiency of national fiscal expenditure follows the same trend as the comprehensive technical efficiency during 2007–2017, with an overall decreasing trend, and starts to increase in 2021, indicating that energy conservation and environmental protection fiscal funds need to be regulated in terms of expenditure scale.

The last column in Table 4 is returned to scale. DRS stands for Decreasing Returns to Scale, which refers to a situation where the rate of increase in output decreases gradually when the inputs increase by a certain percentage. In simple terms, it means that when the scale of environmental protection fiscal expenditure increases, the output will not increase as fast as the input factors. From 2008 to 2020, the efficiency of environmental protection fiscal expenditure in China is decreasing yearly, which indicates that the scale effect of inputs becomes smaller, so it needs to optimize management further and improve efficiency.

It can be seen that, in terms of the degree of effectiveness, most of the years in China are ineffective in terms of comprehensive technical efficiency and scale efficiency. A pure technical efficiency of 1 and a slack variable of 0 indicate that at the current level of technology, the use of China’s environmental financial input resources is efficient. The main factor limiting the improvement of comprehensive technical efficiency is the decline in scale efficiency values, so the focus of improvement is on how to utilize its scale efficiency better.

#### 4.3.5. Summary

The efficiency of fiscal expenditure in China between 2010 and 2019 still had room for improvement, indicating a significant disparity between the government’s utilization of funds and the level of energy conservation and environmental protection work. This suggests poor coordination and integration of energy conservation and environmental policies and planning in previous years. Specific problems include:

Firstly, environmental fiscal expenditure is mainly concentrated on technological transformation and pollution governance, with relatively little expenditure on public health. China’s environmental fiscal expenditure is mostly used for air, water and solid waste treatment. Compared to other areas, such as ecological protection, health risk management and rural environmental improvement, the proportion of environmental pollution governance is higher than required. This weakens the position of public health issues in environmental governance and leads to insufficient coordination and integration between public health and environmental protection.

Secondly, environmental fiscal expenditure needs more special supervision. A regulatory mechanism for environmental protection fiscal expenditure is required to make it easier for the flow of funds to be opaque, duplicate construction, and inefficient use. As a result, the environmental protection financial expenditure on public health protection cannot be realized, making it difficult to bring the actual effect of environmental protection financial expenditure into play.

Thirdly, the execution efficiency of environmental fiscal expenditure could still be improved. The low execution efficiency of environmental fiscal expenditure, combined with a lack of adequate supervision and evaluation mechanisms, results in an inability to effectively evaluate and provide feedback on expenditure effectiveness, as well as to provide effective reference indicators, which affects the overall effectiveness of environmental fiscal expenditure.

Moreover, social participation in environmental fiscal expenditure is below expectation. Environmental fiscal expenditure needs more public participation in the practice process. With insufficient consideration of public opinions and needs, which leads to a lack of universality of the effectiveness of environmental fiscal expenditure, it is difficult to meet the needs of different regions and groups.

Finally, long-term planning for environmental fiscal expenditure is preferred. Lack of such planning may make it difficult to predict and avoid potential environmental risks and public health hazards, as well as scientifically evaluate the long-term benefits of environmental fiscal expenditure, leading to the failure to maximize its long-term effectiveness.

From a public health perspective, China’s environmental protection fiscal expenditure has many problems that should be thoroughly analyzed and improved. In the face of significant differences, China should develop appropriate improvement measures based on its situation instead of blindly copying the experience of countries and regions with relatively high efficiency in energy conservation and environmental protection fiscal expenditure.

## 5. Study on Countermeasures for Optimizing Fiscal Expenditure on Energy Conservation and Environmental Protection in China

This chapter aims to provide targeted optimization suggestions for current issues in China’s energy conservation and environmental protection fiscal expenditure by combining economic theory, objective analysis and empirical results. The recommendations are informed by best practices observed in some foreign countries.

### 5.1. Strengthen Government Supervision and Management of Public Health

Firstly, to improve the financial guarantee system for energy conservation and environmental protection, it is crucial to establish a comprehensive legal system that integrates factors related to health and disease into environmental finance legislation. The guiding ideology of environmental laws should prioritize the protection of public health. Legislative purposes of “protecting public health” in environmental standards should be clarified, and environmental standards should be revised to reflect the core value of caring for people. The legal nature of environmental standards must be defined, and public participation in formulating environmental standards should be encouraged.

Secondly, at the environmental law enforcement level, the responsibilities and authority of various departments should be delineated and unified to prevent the emergence of overlapping law enforcement authority and duplication of law enforcement. The environmental target assessment and accountability system can be implemented to promote strict law enforcement by law enforcement bodies. It is also crucial to organize law enforcement officers to learn relevant business knowledge and improve their personal quality.

Furthermore, environmental administrative supervision must include environment and health in the supervision standards, stipulate specific implementation rules, and require strict implementation by supervisory bodies. Encouraging public participation in environmental administrative supervision is necessary to take full advantage of public involvement’s benefits in environmental management. This will help protect the environment and safeguard public health rights and interests.

### 5.2. Optimize and Adjust the Financial Expenditure System for Energy Conservation and Environmental Protection

China has a significant legacy of environmental pollution, which has led to energy efficiency and environmental protection fiscal expenditure oriented toward environmental issues since its inception. The more complex and severe the environmental and climate problems we face, the more funding is needed to address them. According to international standards, the current scale of China’s fiscal expenditure on energy conservation and environmental protection still needs to be increased. The proportion of GDP invested in energy conservation, and environmental protection led by government finance has only reached the level of curbing the continuous deterioration of the ecological environment. The level of pollution governance and public health satisfaction is difficult to further improve and enhance in this situation, and there is a considerable gap compared with developed countries. Therefore, China should quickly reverse the condition that the growth rate of fiscal expenditure tends to slow down, increase the government’s fiscal expenditure in this field, and eliminate the intervention of local governments. Environmental protection-related departments should also strengthen the punishment of enterprises that emit sewage.

We need to strengthen the government’s fiscal expenditure on energy conservation and environmental protection, improve the fiscal system, and increase the proportion of fiscal expenditure in the government’s public finance expenditure. Research shows that government fiscal expenditure significantly promotes public health. With limited financial resources, reasonable and efficient fiscal expenditures are beneficial to the ongoing assumptions of the fiscal framework and the optimization of the fiscal expenditure structure, which in turn promotes the improvement of public health.

In addition, the structure of fiscal expenditure on energy conservation and environmental protection should be deepened in the economic, social and scientific fields. We have found that as the level of economic development increases, the level of health improvement tends to decrease, and the improvement of health and economic development do not precisely coincide, which means that the improvement of health cannot be achieved simply through economic development. Especially in our country, economic development may be accompanied by environmental pollution and deterioration of the living conditions of residents. Therefore, while maintaining steady economic growth, measures such as environmental management and improvement of public environmental services should be carried out simultaneously.

### 5.3. Establishing a Sound Performance Evaluation System for Financial Expenditure on Energy Conservation and Environmental Protection

Although the status of energy conservation and environmental protection budget expenditure and the way the funds are invested vary from country to country, they all emphasize the performance of the use of funds and provide many practices worth learning.

First, most countries have established a transparent performance evaluation system for environmental financial subsidy programs. For example, the German government has launched a regular evaluation system emphasizing that all financial subsidy programs should be subject to regular assessment regarding goal achievement, efficiency and transparency.

Second, the performance evaluation of specific subsidy projects covers multiple stages, including before, during and after the subsidy, and will take the initiative to publish the performance evaluation results in the form of reports, and the whole process is open and transparent. For example, many countries routinely publish government subsidy reports with the evaluation results of the previous phase of the subsidy program.

Third, it will be based on the performance evaluation of the use of monetary funds for environmental protection, dynamic adjustment of capital investment areas, increase or decrease in the amount of investment, and constantly optimizing the investment program.

Fourth, some countries, represented by the EU countries, have stepped out of the narrow environmentalist perspective when assessing the performance of environmental fiscal funds [43]. Their evaluation indicators not only consider environmental benefits but also tend to include broader economic and social benefits in the assessment system under the framework of green and sustainable development, such as feedback from the public, equity of inclusive growth (the impact of environmental tax policies on low-income people, the impact on small and micro enterprises).

### 5.4. Continuously Strengthen the Driving Effect of Energy Conservation and Environmental Protection Financial Funds on Social Capital

Under the current premise that pollution governance still requires more investment, it is challenging to meet the considerable environmental and public health needs by relying on financial investment alone. Especially in the last two years of the new coronavirus epidemic, many countries are facing the pressure of falling fiscal revenues and rising public expenditure. As a result, governments are emphasizing open source, that is, to strengthen the role of environmental protection monetary funds to guide, leverage and capital synergy and to explore multi-channel investment mechanisms, to improve environmental governance. For example, governments are focusing on cooperation with industries and research institutions. According to the UK Industrial Decarbonization Plan report, effective collaboration between governments, industries and research institutions is crucial in supporting the development of low-carbon technologies at the required scale and speed of advancement. It is important to note that the costs of such initiatives should be shared among the government, businesses and consumers. Moreover, it is crucial to establish a connection with the capital market and innovate using various green financial instruments to engage significant social capital in environmental and public health initiatives with limited fiscal resources. This can improve China’s ecological and environmental governance, as well as public health.

## Figures and Tables

**Table 1 ijerph-20-06018-t001:** Fiscal expenditure on Energy Conservation and Environmental Protection in China, 2007–2021.

Year	Energy Conservation and Environmental Protection Expenditure (Billion RMB)	National General Public Budget Expenditure (Billion RMB)	Share of Environmental Protection Expenditure in Fiscal Expenditure (%)	National GDP
2007	995.82	49,781.35	2.00	270,232.30
2008	1415.36	62,592.66	2.26	319,515.50
2009	1934.04	76,299.93	2.53	349,081.40
2010	2441.98	89,874.16	2.72	413,030.30
2011	2640.98	109,247.79	2.42	489,300.60
2012	2963.46	125,952.97	2.35	540,367.40
2013	3435.15	140,212.10	2.45	595,244.40
2014	3815.64	151,785.56	2.51	643,974.00
2015	4802.89	175,877.77	2.73	689,052.10
2016	4734.82	187,755.21	2.52	743,585.50
2017	5617.33	203,085.49	2.77	827,121.70
2018	6297.61	220,906.00	2.85	919,281.10
2019	7390.20	238,874.00	3.09	990,865.10
2020	6333.40	245,588.00	2.58	1015,986.20
2021	5525.14	246,322.00	2.24	1143,669.70

Date from the Statistical Yearbook.

**Table 2 ijerph-20-06018-t002:** Key areas of central and local expenditure on energy conservation and environmental protection in 2021.

Central		Local	
Expenditure Items	Proportion of Expenditure (%)	Expenditure Items	Proportion of Expenditure (%)
Energy management services	40.54	Pollution prevention	37.60
Other energy conservation and environmental protection expenditure	31.05	Other energy conservation and environmental protection expenditure	17.99
Renewable energy	7.05	Natural ecological protection	11.36
Natural forest protection	6.40	Energy conservation and utilization	9.63
Pollution reduction	5.07	Environmental protection management services	6.39
Pollution prevention	2.13	Pollution reduction	6.30
Environmental protection management services	2.09	Natural forest protection	3.85
Environmental monitoring and surveillance	1.99	Returning farmland to forest	2.63
Natural ecological protection	1.71	Environmental monitoring and surveillance	1.29
Energy conservation and utilization	1.59	Energy management services	1.01
Returning farmland to forest	0.33	Circular economy	0.77
Returning grazing to grass	0.04	Renewable energy	0.71
Circular economy	0.01	Returning grazing to grass	0.24
		Wind and sand desert management	0.16

Data from the Statistical Yearbook.

**Table 3 ijerph-20-06018-t003:** Input and output indicators.

Projects	Category		Specific Indicators	Unit
Inputs	Energy saving and environmental protection input		Per capita financial expenditure on energy conservation and environmental protection	RMB/person
Expenses	Pollution governance	Expected output	Centralized treatment rate of sewage treatment plants	%
			Harmless disposal rate of domestic waste	%
		Non-desired outputs	Industrial smoke (dust) emissions	ton
			Industrial sulfur dioxide emissions	ton
			Total industrial wastewater discharge	million tons
	Energy savings	Expected output	General industrial solid waste comprehensive utilization rate	%
		Non-desired outputs	Industrial electricity consumption per unit of GDP	kWh/million RMB
			Urban water supply per unit of GDP	Tons/million
	Ecological restoration	Expected output	Greening coverage of built-up areas	%
			Artificial forestation area	hectares

**Table 4 ijerph-20-06018-t004:** DEA efficiency values for China, 2007–2021.

Year	Comprehensive Efficiency	Pure Technical Efficiency	Scale Efficiency	Compensation for Size
2007	1	1	1	-
2008	0.952	1	0.952	drs
2009	0.789	1	0.789	drs
2010	0.625	1	0.625	drs
2011	0.682	1	0.682	drs
2012	0.664	1	0.664	drs
2013	0.618	1	0.618	drs
2014	0.595	1	0.595	drs
2015	0.573	1	0.573	drs
2016	0.606	1	0.606	drs
2017	0.631	1	0.631	drs
2018	0.635	1	0.635	drs
2019	0.591	1	0.591	drs
2020	0.798	1	0.798	drs
2021	1	1	1	-

## Data Availability

The data in this study are publicly available.

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
