# Peer review of "From the Perspectives of Pollution Governance and Public Health: A Research of China’s Fiscal Expenditure on Energy Conservation and Environmental Protection"

_ijerph, 2023, doi:10.3390/ijerph20116018_

Round 1

Reviewer 1 Report

The topic of the article is interesting. Some suggestions:

It should be better justified why the topic is very relevant today with additional literature

The theoretical background could be enriched. Likewise, the results should be supported by scientific evidence. 

The paper is very simple in understanding, it should add some insights for future research. 

It is not very clear what conclusion the paper should reach, as the sub-chapters have very long titles that confuse the reader. 

Reviewer 2 Report

The article deals with a relevant topic, the text presents an adequate structure and makes an initial approach on the issue of Chinese government investment in an attempt to change the environmental scenario in the country. However, the research method needs to be clarified, especially with regard to the definition of the indicators used in the research. Regarding the discussion of the results, I missed a greater citation of other studies that could corroborate the results found, as it becomes scientifically important for the results to be ratified, so as not to remain only in the opinion of the authors. As well as it is necessary to improve the conclusion of the research.

Reviewer 3 Report

Dear Authors,

the topic you have taken up is interesting and worthy of attention. However, in order for the work to take on a fuller shape and greater scientific value, the literature review section should be expanded, the disussion section, which compares the research you have done with what has already been achieved in this field, is also missing, and there is a more detailed description of the research gap. 

You have only 21 items in the literature - this should be expanded at least twice. 

I hope my comments will be helpful. 

Reviewer 4 Report

Interesting and timely article.

1. Make the abstract readable: The abstract should be refined: lack of a precise description of the research problem, hypothesis, as well as the research tool that was used for the research.
objective of the work,
research problem,
hypothesis, z
the research method used and
basic conclusion;

2. Specify the description of the research methodology,

3. There is no information whether research on this subject has been conducted before

4. Correct the conclusion, because the written conclusions are too general
Research results - carried out correctly, although the research methodology should be described in more detail.

5. Update the bibliography and adapt it to the current problems raised by the authors

In the summary, the conclusions of the research should be clearly articulated.

Round 2

Reviewer 1 Report

Minor integrations in English language are required. 

Author Response

I have done some integration and modification in the English language.

Reviewer 2 Report

After revision, the article is more suitable for publication, with minor adjustments to be made regarding references and spelling throughout the text.

Author Response

I have made some adjustments and changes to the regarding references and spelling.